# The Non-Linear Impact of Financial Development on Environmental Quality and Sustainability: Evidence from G7 Countries

**DOI:** 10.3390/ijerph19148382

**Published:** 2022-07-08

**Authors:** Cristina Ruza, Raquel Caro-Carretero

**Affiliations:** 1Applied Economics Department, Economics and Business Faculty, National Distance Teaching University (UNED), C/Senda del Rey no. 11, 28040 Madrid, Spain; 2The AON Spain Foundation Chair in Disasters, The University Institute of Studies on Migration, Comillas Pontifical University, C/Alberto Aguilera 23, 28015 Madrid, Spain; rcaro@comillas.edu

**Keywords:** CO_2_, greenhouse gases, methane, nitrous oxide, ecological footprint, environmental sustainability, financial development, Environmental Kuznets Curve

## Abstract

This paper analyses the impact of financial development on the environmental quality and sustainability for the group of G7 countries over the period 1990–2019 based on static panel data-fixed effect models. The objective is to explore if there exists a non-linear relationship between the whole financial system development and a wide array of measures of environmental sustainability and degradation, namely adjusted net savings, greenhouse gas, CO_2_, methane, nitrous oxide emissions and ecological footprint. We define a new Financial Environmental Kuznets Curve (FEKC) by introducing the square term of financial development on the environment-finance relationship. Empirical results prove the existence of non-linear relationships between the composite index of financial development and environmental degradation for the group of advanced economies. In the case of methane, we validate the presence of an inverted-U shape association in line with the FEKC hypothesis, while for greenhouse gas and CO_2_ the link follows a U-shaped pattern. The impact of financial development on environmental sustainability is monotonically positive and statistically significant while the ecological footprint is not statistically linked with the level of financial development within G7 countries. Economic growth, human capital, population density and primary energy consumption appear as significant drivers of environmental quality and sustainability.

## 1. Introduction

Over the last decade increasing concerns have been arising in the fight against climate change, global warming and biosystems’ degradation. Since the approval of the Kyoto Protocol different commitments have been assumed by countries worldwide and nowadays environmental protection is at the core of policymakers’ agenda.

A decisive step from an international point of view is the commitment of well-developed countries to the achievement of Sustainable Development Goals (SDG). This initiative requires important financial support from public authorities and private financial systems because the transition process would not be possible if enough financing is not available for changing the prevailing economic model and promoting the more pollutant sectors to evolve towards a neutral carbon economy by 2050.

The specialized literature apprehends much attention to the negative externalities associated with environmental damage and climate change, and it has involved diverse disciplines like ecology, economy, and law, just to cite few [1].

In particular, the role of the financial system has been widely analyzed in the literature from different perspectives, like its impact on the level of economic growth, technological progress, and income inequality [2]. The primary objective of a well-developed financial system is to fulfil the basic needs of funds channeling, support of the payment system and the provision of financial services. Once these minimum standards have been achieved, it is expected that financial systems evolve in line with economic growth and foster economic development and prosperity.

However, until recent days prosperity has only been measured in socioeconomic terms, but with no reference to the natural environment. Quite recently for the sake of measuring prosperity new metrics aligned with ESG criteria (environmental, social and governance) are being used. However, our current economic model focused on a linear approach and based on increasing industrialization and commercialization demands high energy that results in high emissions and a serious threat to human beings [3]. The effects of the environmental degradation are not restricted to the economic sphere. According to [4] “the prevailing global warming and the subsequent climate change pose potential diverse physical, ecological and health threats reciprocated by extreme weather conditions”. Indeed, the environment is closely related to human health issues because there is a direct effect of pollutant emissions on a varied range of cardiopulmonary diseases and child mortality, not to mention problems associated with water quality and scarcity [5,6]. Recently, some authors have explored the link between air quality and the coronavirus disease rapid spread [7]. We should not ignore that all these harmful effects are accompanied by important economic costs of higher medical expenditures, lower labor productivity and losses of human lives. The OECD publishes specific data on mortality, morbidity and welfare costs from exposure to environmental-related risks.

Having emphasized the importance of the financial system as facilitating the transition towards a carbon neutral economy and a sustainable development model, this study sheds new light on the linear and non-linear impact of financial development in terms of environmental degradation and sustainability. In this paper we define a new Financial Environmental Kuznets Curve hypothesis (FEKC) as the existence of an inverted U-shaped relationship between levels of financial system development and levels of environmental degradation.

We will analyze the group of G7 countries that are financially developed and well positioned to guide developing countries within the path towards sustainability because if developed countries do not take decisive steps in these years, global goals of planet sustainability would not be achieved. [8] argued that developed countries have better ability to climate change adaptation than developing countries.

In particular, the article will explore if there is a threshold after which the level of financial development exerts a positive impact on the environment, either by reducing polluting emissions or increasing levels of environmental sustainability. When countries financially evolve and adopt greener technologies, switch to a more intensive use of renewable energies or invest more heavily in research and development activities, this could result in diminishing polluting emissions. This seems an interesting topic that deserves to be further empirically examined if we are aimed at simultaneously achieving an effective environmental protection and ambitious sustainability standards in the medium term.

Specifically, this paper set the following research objectives:(1)To assess which are the drivers of environmental burden and sustainability for the specific group of developed countries.(2)To analyze to what extent developed financial systems are well positioned to protect the environment and help reducing polluting emissions.(3)To assess the existing nexus between financial development and different proxies of environmental degradation at different stages of development, and explore if there is a U-shaped relationship (the so-called Financial Environmental Kuznets Curve, FEKC).(4)To give some recommendations for polluting emissions’ abatement strategies based on the empirical findings of this study.

In this paper we adopt a panel data-fixed effect analysis that explains time-invariant country-specific features that may create omitted-variable bias. We also include a wide range of environmental damage variables because there is evidence that they are driven by different forces, to a different extent and in different directions [9]. We will analyze global emissions of greenhouse gases (GHG, hereafter), its three main components, namely carbon dioxide (CO_2_), methane and nitrous oxide emissions, ecological footprint and a proxy of environmental sustainability.

The uniqueness of the paper lies on analyzing the non-linear relationship between financial system development and the natural environment within the EKC framework, assuming that this link is non-permanent, and it depends on the country’s phase of financial development. To the best of our knowledge no previous articles have particularly tested the non-linear impact of financial system development for the specific group of advanced economies and include into the analysis so diverse measures of environmental quality and sustainability as this study. Secondly, to assess the importance of the financial system we use a composite index of financial development that captures both the intermediation activity of banking institutions and the capitalization process carried out through capital markets, which gives a wider perspective of the financialization process than previous studies that are only focused on the credit provision channel [10]. Third, instead of considering levels of CO_2_ emissions, this study also includes varied environmental variables like the ecological footprint, the three main GHG (CO_2_, methane and nitrous oxide emissions) and also the environmental sustainability measured by the adjusted net savings. Fourth, new variables like the expenditure on research and development activities and a human capital index will be also appraised in this setting. Finally, this article extends previous studies by using more recent data so our findings contribute to the open debate and results can be contrasted with past studies. The data base and the open-source code required to replicate all analyses in this article (including multicollinearity diagnoses, descriptive and bivariate correlations procedures) are available in [11].

The remainder of this paper article is organized as follows. The next section reviews the literature on the topic and summarizes the main results achieved by previous studies so far. The section of data and empirical model describes the sample and the econometric technique applied. The fourth section presents the main empirical results, and the implications of the findings are discussed in the fifth section. The article ends with some concluding remarks that outline some recommendations for policymakers and regulators.

## 2. Literature Review

The existing nexus between the process of financialization and economic growth has been extensively analyzed in the literature. The origin of this line of research dates back to the study of [12] that defines a model for economic growth and its short-term and long-term equilibriums.

The link between the environment and economic growth has been hypothesized in the Environmental Kuznets Curve (EKC) framework. The EKC phenomenon was first established in the pioneering work of [13], which proves the existence of an inverted U-shaped relationship between growth and environmental quality. According to the EKC hypothesis, at the initial stage of growth a rise in income per capita causes high emissions and has a negative effect on the environment, but after reaching a certain threshold level, further increases in income reduces CO_2_ emissions and has a positive effect on environmental quality.

Other line of research focuses on the link between growth, financial development and CO_2_ emissions. From a theoretical point of view [14] have identified different channels through which the financial system may have an impact on the natural environment, which are capitalization effect, technology effect, income effect and regulation effect. The sign of the relationship will ultimately depend on which of the previous effect is dominant.

Empirical studies carried out reveal that results are sensible to the choice of methodology, the sample of countries, the set of explanatory variables and the period of analysis considered [15,16]. Thus, no consensus has been yet reached on this topic. Indeed, most of studies have been mainly focused on the effects of economic growth and financial development, among other drivers, on the level of CO_2_ emissions.

From the literature reviewed in this study we distinguish four groups of studies. Within the first group, some authors like [17] encounter a positive relationship between financial development and environmental quality because financial development helps in providing higher information about the importance of the environment, especially in developing countries. They find that foreign direct investment contributes to diminishing levels of CO_2_ emissions per capita, while the financial liberalization effect will ultimately depend on the strength of the institutional framework in force. In addition, these studies support that financial sector appears to be providing financial services for eco-friendly programs at a lower cost and are specialized intermediaries in financing these types of programs. Ref. [18] explore the relationship between financial development and environmental damage and argue that it is not significant in low-income countries because of their early stage of economic growth. However, for the group of upper-medium income countries, the outcome is the opposite and financial development enhances environmental quality. One explanation is that these developed countries have well-established financial systems that positively correlate with economic progress and financial systems are less intensive in capital than industrial production, thus generates lower CO_2_ emissions. Ref. [19] reveal that financial development, urban population and technology ensure an improved environmental quality in the long run in emerging economies, but in the short term they encounter a bidirectional causal relationship. Ref. [20] analyzes the linkage between financial development and the reduction of CO_2_ emissions related to a level of income inequality that should not be exceeded in order to maintain this effect. Ref. [21] explore the impact of domestic credit to the private sector subject to the level of trade by using fixed effect panel threshold model in the BRICS economies and find that environmental quality increases consistently across all intervals.

Within the second group of studies, ref. [22] analyses the positive relationship between polluting emissions and economic growth. Ref. [23] argues that financial development facilitates the access to credit for setting up businesses that are intensive in energy consumption, therefore increasing environmental burden. Ref. [24] supports that financial development reduces transaction costs and makes credits to the private sector relatively cheaper. This leads to the undertaking of new projects and buying new facilities that in the end will upsurge polluting emissions. Ref. [25] points out that industrial activities generally accompany financial development, which in turn have negative externalities of increasing pollution levels. A great deal of studies has focused on the group of developing countries due to their specific characteristics. Refs. [16,26,27,28,29] find a direct effect of economic growth in terms of increasing environmental degradation. However, scarce attention has been paid to the group of developed economies as the more pollutant cases. Among them, ref. [30] discover a positive relationship between credit provided by banks and GDP, and indirectly with CO_2_ emissions, while [28] find a direct and positive effect of financial development on CO_2_ levels for the group of G8 countries, although this effect is even more pronounced for the group of D8 countries.

A third group of papers is characterized by mixed results when analyzing an extensive panel of countries [31,32,33] from which no conclusive results can be obtained.

Finally, a fourth group of studies do not encounter a significant relationship between financial development and environmental quality like [34,35,36,37].

Table 1 below summarizes some important contributions to this field of study.

However, the empirical debate goes beyond the linear association of income, financial development and the environment, and some authors have explored this link within the Environmental Kuznets Curve (EKC) framework based on the work of [41] that analyses the non-linear empirical connection between economic growth and environmental quality. Empirical findings reveal the presence of an inverted U-shaped curve suggesting that income increases initially leads to higher levels of polluting emissions, but after a level the negative impact turns into positive by reducing environmental damage. Therefore, countries growing beyond this threshold can be considered as positive for the natural environment [27]. The turning point can be interpreted as the consequence of advances towards a more efficient use of energy and the implementation of environmental protection initiatives. According to [9] the theoretical explanations of this finding are three-fold. First, the variation of marginal utilities of economic growth and environmental quality, implying that when a country’s income grows then the rate of return of reducing pollution tends to increase. Secondly, the “pollution haven” effect that explains the relocation of most pollutant industries from developed to developing countries as a sort of environmental dumping, therefore reducing environmental damage. Last, a sectorial recomposition in favor of environmental- friendly activities that alleviates pollution. In addition, [33] point out that as people disposes of extra income this makes them be more natural environmentally oriented.

Within the EKC framework a great body of specialized literature has analyzed the level of carbon emissions [35,42,43,44,45], but ignore other emissions that can significantly harm the environment [46]. Other lines of research tend to focus on alternative measures of environmental damage like [47,48] who analyze GHG emissions, [49,50] study the ecological footprint and [33] includes into the analysis a varied mix of environmental quality variables and environmental sustainability. In the same vein, [51] focus on Islamic countries and [9] on the group of EU countries and explore the three main GHG gases. The reasoning is that environmental quality cannot be captured by levels of CO_2_ while ignoring degradation in soil stock, forestry stock, mining stock or oil stock, among others.

If we specifically analyze the evidence of EKC on developed countries [52,53] find an inverted U-shaped relationship in the case of France, and [48] in the US. Ref. [37] analyze the case of the 10 top-ten emitter countries applying quantile regressions and the findings of the study validates the EKC hypothesis only in top quantiles. Ref. [54] performs panel data analysis and find support of the EKC in American and European countries at all quantiles. Ref. [15] apply dynamic seemingly unrelated regression long-run panel and their results support the validity of the EKC hypothesis for 5 of the 18 Central and Eastern European countries, so partial support is empirically demonstrated. Likewise, ref. [18] conclude that among a mixed panel of countries the EKC only holds for the group of developed countries, in line with previous findings of [55]. On the contrary, ref. [49] do not support the EKC hypothesis for the group of EU countries applying a second generation of panel data analysis. Ref. [9] validate the existence of a U-shaped relationship for all environmental variables considered in their study for EU-27 and EU-15 countries in the short-run.

Another line of research tries to identify a potential N-shape relationship between the environment and economic growth by including a square and cubic term into the equation. For instance, ref. [1] separately analyze three regions in China and they estimate two inflection points of the inverted N-shaped model for CO_2_ emissions. Ref. [56] confirm the inverted N-shaped relationship between the pollution factor and economic growth per capita in China at a province level. Ref. [57] reveal that GDP per capita has an inverted N-shaped impact on environmental deterioration, having the financial development a direct and moderating effect. Within the same line [58] support the presence of an inverted N type after adding spatial effects. Ref. [59] find a cubic relationship of economic growth and reveal that different renewable and non-renewable energy proxies in Egypt follow a N-shape pattern.

However, up to this point, there is a gap in the existing literature because the non-linear impact of financial development has not been considered within the EKC framework for the group of advances economies. Indeed, we find just few studies that deal with this issue, like [60] at a province level in China and they conclude that initially financial development exerts a positive effect on the environment due to the technological effect (energy efficiency improvements). However, after a certain level additional increases of financial development lead to augmenting environmental damage (U-shaped form). Ref. [61] find the opposite outcome of an inverted U-association, indicating that at a second stage of economic growth financial development becomes environmentally friendly in the presence of strong economic institutions.

In the light of this mixed and inconclusive evidence we argue that there remains room for exploring the non-linear impact of financial development on environmental quality and sustainability in advanced economies. In this paper we define the Financial Environmental Kuznets Curve hypothesis (FEKC) as the existence of an inverted U-shaped connection between levels of financial development and environmental degradation, so as long as the financial system develops after a threshold the natural environment will benefit from technological progress, greener technologies, move to renewable sources of energy and the implementation of initiatives that help reducing existing levels of polluting emissions. This hypothesis should follow a U pattern in the case of environmental sustainability.

## 3. Data and Empirical Model

This section analyses the data and the econometric strategy applied in this research.

### 3.1. Data

This study covers annual time series data from 1990 to 2019 for the group of G7 countries (Canada, France, Germany, Italy, Japan, the UK, and the USA).

These countries have been selected because their levels of economic growth are the highest worldwide, so it can be expected that their financial systems are also well-developed. The aim of this study is to capture the effect of financialization at a second stage of development to see if non-linear patterns appear in this relationship.

The selection of variables is based on existing literature and focused on the specific group of advanced economies. In this vein, a wide array of environmental proxies have been included to extend previous studies and simultaneously analyze global GHG emissions and its components, namely CO_2_, Methane (MET) and Nitrous oxide (NIT) (Time series of polluting emissions are included in Appendix A). In addition, the Ecological Footprint (EF) has been included in order to take into account the biosystems’ degradation. Finally, a proxy of environmental sustainability has been considered as the variable Adjusted Net Savings (ANS).

In this background we assume that environmental quality and sustainability are a function of important variables such as the GDP and the level of Financial Development (FD). Other important drivers of this relationship that should be controlled in advanced economies are Population Density (POP), Expenditure on Research and Development (RD), Primary Energy Consumption (PE) and Human Capital (HC). Each of these variables play an important role in advanced economies and exert a direct impact in environmental damage and environmental sustainability as argued in the literature review section.

The definition and data sources from official statistics of selected variables are shown in Table 2, while the descriptive statistics (i.e., mean value, standard deviation, maximum value, and minimum value) of all the selected variables are shown in Table 3. Research data was gathered from the World Bank database, International Monetary Fund database, CAIT Climate Data Explorer, Our World in data, Penn tables and the Global Carbon Project.

### 3.2. Estimation Model

This section outlines the econometric instrument deployed in the study. In order to empirically explore the impact of financial development on the environmental degradation and sustainability econometric model panel data regression with fixed effects is used for the baseline model. The term “fixed effects” is due to the fact that, although the intercept may differ across individuals (the 7 countries), each individual’s intercept is time invariant [62,63].

Ref. [64] suggest that panel data estimation models have several advantages over time series data, such as it provides robust results and counters the issue of multicollinearity, heterogeneity and endogeneity. However, fixed effect regression accounts for unobserved time-invariant among individual characteristics, and that may lead to biased results.

To overcome this issue [65] employed the system generalized method of moments (SYS-GMM) to estimate a dynamic panel model that eliminates the countries’ specific heterogeneity by using the first difference of the dependent variable. However, while using cross-country data, we have a number of reasons to use a fixed effect model. First, we assume that those time-invariant features are unique to the country and may not be correlated with other country’s characteristics. Each country is different and therefore individual country’s error term and the constant (which captures individual country characteristics) may not be correlated with the others. Secondly, we assume that something within each country may impact or bias the predictor such as GDP or carbon emission, the outcome variables. This potential effect is unobservable; however, we can control it by using a fixed effect model. In this regard [4] propose to use fixed effect model instead of the model with random effects. The latter is more efficient (the variance of the estimation is lower) but less consistent than the fixed effects model, i.e., it is more accurate in the calculation of the parameter value but it may be more biased than the fixed effects model.

This study considers five important proxies of environmental quality (ENV): GHG, CO_2_, MET, NIT and EF. Additionally, we include the variable ANS as a proxy of environmental sustainability [33].

In this paper we use as the proxy of financial development the composite index of financial institutions and financial markets development in terms of depth, access and efficiency. For alleviating omitted variable bias, we sequentially added several control variables that are possibly connected with variation in environmental quality and sustainability. In the selection of control variables, after referring to the existing literature, we decided to control four variables, including Population Density (POP), Expenditure on Research and Development (RD), Primary Energy Consumption (PE) and Human Capital (HC).

Therefore, we can express the association of ENV with economic growth, financial development, population density, expenditure on research and development, primary energy consumption and human capital as follows:ENV = f (GDP; FD; POP, RD, PE, HC),(1)

All variables in Equation (1) are transformed into their natural logarithms to eliminate the effect of variable dimension as well as to reduce dispersion in the data and to minimize issues related to potential multicollinearity and heteroscedasticity in the data. The loglinear transformation data also produce more efficient and consistent results than the simple linear form [66]. The log-linear multivariable model is shown as follows:(2)$$LogENV_{it}=\beta_{0}+\beta_{1}LogGDP_{it}+\beta_{2}LogFD_{it}+\beta_{3}LogPOP_{it}+\;\beta_{4}Log\;RD_{it}+\beta_{5}LogPE_{it\;\;}+\beta_{6}LogHC_{it\;}+\varepsilon_{it}$$
where, *i* denotes the country (*i* = 1,…,7), and t indicates the time period (1990–2019). Log ENV appraises environmental variables. The $\beta_{1},$ $\beta_{2},$ $\beta_{3}\;,\beta_{4}\;,\beta_{5}\;\mathrm{and}\;\beta_{6}$ coefficients correspond to GDP, FD, POP, RD, PE, HC, respectively, and the parameters can be interpreted as elasticities of ENV with respect to these variables. $\varepsilon_{it}$ denotes the error term.

Based on previous studies that found a non-linear relationship between GDP and environmental quality (the so-called Environmental Kuznets Curve), this study goes further by analyzing the non-linear nexus between ENV (quality and sustainability) and FD to investigate the long-run relationship. We test if there is an inverted U-shaped association indicating that at the early stage of financial development the natural environment deteriorates as the financial system evolve, but when achieving a particular threshold of financialization the level of environmental degradation begins to fall. To test this hypothesis, we add the FD- squared value to test the validity of the so-called Financial Environmental Kuznets Curve (FEKC) hypothesis. In regressions we also account for the fact that financial development is correlated with economic growth, and so the former may simply pick up the effect of a general increase in wealth on the level of energy demand. We therefore add both FD and the square thereof to the regression. We rewrite the general model as follows:(3)$$LogENV_{it}=\beta_{0}+\beta_{1}LogGDP_{it}+\beta_{2}LogFD_{it}+\beta_{3}LogFD_{it}^{2}+\beta_{4}LogPOP_{it}+\beta_{5}Log\;RD_{it}+\beta_{6}LogPE_{it\;\;}+\beta_{7}LogHC_{it\;\;}+\varepsilon_{it}$$

According to the model, if the FEKC hypothesis defined in this article holds, then *β*_2_ should be statistically significant and positive while *β*_3_ would appear as statistically significant and with a negative sign. In this vein, an inverted U-shaped relationship between financial development and environmental degradation would be validated.

In the case of environmental sustainability, the same outcome will be obtained when the relationship follows a U pattern, then the effect of initial financial development will be negative by reducing environmental sustainability up to a point, and thereafter sustainability will start to increase along with financial development.

Table 4 shows the outcome of the pairwise correlation matrix. It reveals a positive correlation between FD and GHG, CO_2_, MET, NIT and EF. Also, HC, POP, RD and PE show a positive correlation with emissions. The correlation between FD or GDP and ANS is negative.

## 4. Empirical Results

Next tables present the outcomes of panel fixed-effect regression for each of the 6 dependent variables defined in this study. Regarding the assumptions of the regression, the Durbin-Watson statistics suggests that there is no autocorrelation in the data. The test of normality indicates that we can accept the normality assumption. The findings depict that the overall panel data regression models with fixed effects are good (except for the variable ANS adjusted R^2^ > 0.8).

The per capita CO_2_ emissions were hypothesized to be related to the level of economic development (proxied by per capita GDP) and financial development (FD) following [26]. In fact, the carbon emissions in a country do not necessarily depend on its income level alone; financial development may be another source. In order to evaluate this, goodness of fit tests [67] have been carried out to compare the adequacy of the models with or without the variable GDP. Those models with both variables are preferred according to the adjusted R-squared, AIC-Akaike information criterion, BIC-Bayesian information criterion- and Log-likelihood (see Table 5, Table 6, Table 7, Table 8, Table 9 and Table 10). The objective of model selection is to estimate the information loss when the probability distribution associated with the true (generating) model is approximated by probability distribution associated with the model that is to be evaluated.

Table 5 shows the fixed effect panel regression results of the association between FD and overall GHG. First, the linear model reveals that FD and PE exerts a monotonic and positive impact on the overall level of GHG emissions, while POP and HC counteract this effect by diminishing pollution levels. In this model GDP and RD appear as no significant with *p*-values higher than 0,1. The previous study of [68] found a bidirectional Granger causality link between economic growth, energy consumption and GHG for a group of 16 Asian countries.

Secondly, the non-linear (quadratic) model seems to better capture the behavior of GHG and almost all variables appear significant. In this vein, we find that GDP, POP and HC are contributing to reduce GHG harmful emissions, while PE creates a significantly positive link with GHG. Conversely, [48] analyses GHG emissions at a sector level in the US and confirms the validity of the EKC hypothesis (inverted U shape) for the specific relationship of economic growth and GHG, so only after the turning point the effect of GDP would turn into negative. Of particular interest in this quadratic model is the composite index of financial development, which is statistically significant and negatively linked to GHG indicating that a 1% increase in FD will reduce GHG emissions by 9.1%, signifying a mitigating effect of FD on environmental degradation. Meanwhile, the coefficient of FD square is positive and significant at 1%. These results demonstrate the presence of a U-shaped association between environmental damage and FD for these sampled countries. We can argue that FD is not accompanied by reduced levels of GHG as should be expected. Indeed, the beneficial effect on the environment reaches a minimum point after which environmental degradation starts to increase.

The next step of our research procedure consists of individually analyzing the three main components of GHG, namely CO_2_, NIT and MET. Results of the panel data-fixed effect model of CO_2_ emissions are presented in Table 6. Without taking the magnitude of coefficients into account, the linear model of CO_2_ reveal that PE outlines a statistically significant positive connection with carbon emissions, while HC develops a negative and significant impact on CO_2_. This study also demonstrates that there is a monotonically heightening (positive) association between FD and CO_2_. This result is in line with previous results of [69] for 21 North American economies and [61] who analyze an extensive panel of more than 100 countries. These authors show the positive impact of financial development on CO_2_ emissions, and that this effect is reduced when including institutional factors into the analysis. The study of [15] specifically considered the case of Eastern and Central European countries, and they find that financial development helps to reduce CO_2_, while energy consumption is the key determinant of CO_2_ emissions in line with the empirical results presented. The recent study of [70] also demonstrates that energy consumption leads to higher carbon emissions. In this study, the rest of variables like GDP and RD appear not significant in the linear model of CO_2_.

The empirical findings of the non-linear model support that FD is statistically significant and negatively associated with CO_2_. Howbeit, the squared variable of (FD) is significantly positive so these findings suggest a U-shaped connection between the composite index of financial development and carbon emissions in these countries. These results are aligned with those obtained for GHG, hence FD is not contributing to the de-carbonization of countries after a threshold level (the FEKC does not find empirical support). To the contrary, ref. [21] applied panel threshold models and find that domestic credit to the private sector develops a significantly negative association with carbon emissions but with different intensity depending on the interval of trade considered, so the impact is not equal, but it is negative in all the three intervals of trade defined. Moreover, results of the quadratic model prove that HC is currently empowered to mitigate emissions while PE is one of the major causes of carbon level increases. Likewise, ref. [35] demonstrate that energy had led to high CO_2_ emissions in the US over the last fifty years. Ref. [28] using the ARDL technique for the group of G8 and D8 separately reveal the positive effect of energy use on the augmentation of environmental degradation in both groups of countries.

Results on NIT as the proxy of environmental degradation are presented in Table 7. The linear model outlines that GDP, POP and RD are all significant drivers and exhibit a monotonically negative connection with this local pollutant. The variable FD appears as not statistically significant. Other studies like [9] find a U-shaped connection between nitrous oxide levels and GDP in the short run for the group of EU countries applying panel grey incidence analysis, even though results are sensitive to the model adopted. [56] analyze ammonia nitrogen emissions in Chinese provinces and results reveal a N-shape relationship with economic growth per capita.

In the quadratic model the results remain the same and GDP, POP and RD contribute to reducing levels of NIT. We can conclude that FD does not have a significant impact on NIT, mainly caused by agricultural and soil management activities. Moreover, HC and PE are also non-significant in this setting.

The third pollutant analyzed in this study is MET (Table 8). Empirical results of the linear model show that GDP, POP and RD are significantly helping reduce existing levels of MET, while HC is statistically significant and positively linked. In their study [71] use a proxy of human capital based on the number of patents instead of the level of education and find the opposite outcome.

In the quadratic model the picture appears more complete because a squared term of FD is introduced for appraising the non-linear effect. In this case, a 1% increase in FD is spurring MET (positive sign). The squared estimate of FD is significantly negative, hence for these sampled economies there is an inverted U-shaped association between FD and MET. In the light of these results, the FEKC hypothesis is validated. The rest of variables of the squared model present the same signs as the linear model, so GDP, POP and RD develop a significantly negative relationship with MET. In this case HC is the driving force behind increases in MET. A recent study of [72] applied panel Granger non-causality test and discovered a bidirectional causality link between methane emissions and economic growth. What is more, the EKC hypothesis of an inverted U-pattern relationship of growth and methane emissions holds for the group of CEMAC countries (Central African Economic and Monetary Community). [51] present evidence in Islamic countries in favor of non-linear patterns in environmental quality indicators related to economic growth. Methane, ecological footprint and CO_2_ follow an inverted U-shaped pattern.

The next variable to be analyzed is EF (Table 9). Under the linear specification, the findings show that GDP generates a statistically significant positive effect on EF, whereas HC is monotonically negative in this relationship. The same result was achieved by [73] for the case of developing countries in which human capital presents a negative association in the long term.

The quadratic model confirms these results and that FD is non-statistically significant for explaining EF in advanced economies. On the contrary, ref. [18] reveal in their study that FD helps reducing EF in high income countries.

Finally, for appraising environmental sustainability in this study we consider the variable ANS, and the results are show in Table 10. Empirical findings support that GDP and FD are significant drivers exerting a positive impact on environmental sustainability in line with previous findings of [33] for a group of OECD countries that validates the presence of an inverted-U relationship between environmental sustainability and GDP. The results of this study also show that POP, HC and RD produce a significantly negative effect on existing levels of environmental sustainability.

In the quadratic model the same results are confirmed except for RD that appears as no significant in this setting.

## 5. Discussion

Within the EKC framework the results of this study make significant contributions to this area of research by adding new evidence of the linear and non-linear impact of the whole financial system on the natural environment. There is a gap in the existing literature because the role of the financial system has been basically analyzed within the domain of linear models and focusing on the channel of credit provision. In this study it has been demonstrated that when including a quadratic term of financial development new results arise, so the real nature of this relationship can be better appraised. Until recently the impact of financial development has been classified as positive, negative or non-significant. This study provides empirical evidence demonstrating that this relationship is non-permanent, and it evolves with the country’s phase of financial development.

In addition, financial development has been appraised by simultaneously including the development of financial intermediaries, like banks providing credits to the private system, and the development of financial markets as liquidity providers. Only by jointly analyzing these two pillars of financial systems it would be possible to deeply comprehend the overall impact of the financial system and extract some useful conclusions.

Empirical findings show that the sign of the relationship between financial system development and GHG and CO_2_ emissions change from the linear to the non-linear models. Under linear specifications results prove that financial systems are contributing to increasing levels of GHG and CO_2_, but when introducing a non-linear term, this relationship becomes non-linear and follow a U-shaped form.

What is more, it seems vital to perform a disaggregated analysis of the three main GHG, whose behaviors are markedly different as it has been showed in this study and aligned with previous studies. Findings reveal that the overall impact of financial system development on GHG emissions is the net result of positive and negative impacts on its components, and these should be separately analyzed. This study identifies the presence of a U-shaped relationship between financial development and carbon emissions, while the opposite outcome of an inverted U-shaped pattern is identified in the case of methane emissions (the FEKC hypothesis).

These days increasing attention is being paid to the relevance of methane emissions due to its properties and potential for reducing CO_2_ levels in the long run. This is a consequence of the shorter period of oxidation of methane gas than that of carbon emissions. It is estimated in 10 years the period of oxidation after which methane molecules will be transformed into CO_2_, so having a warming potential 28 times higher than CO_2_ [71]. Thus, any effort made by current generations in reducing methane emissions will render positive and visible results in the medium term.

Across all models analyzed in this study some interesting conclusions can be drawn. First, primary energy consumption is one of the major forces behind increasing pollution levels in advanced economies. It seems imperative to advance towards a new economic model more reliable on clean sources of energy and that simultaneously help reduce countries’ energy dependence on fossil fuel energies. Additionally, any improvement in energy efficiency and the promotion of high-tech innovations can help reduce energy intensity levels.

Secondly, economic growth measured by GDP per capita exerts a positive effect in terms of environmental protection by reducing GHG, methane and nitrous oxide emissions as well as increasing environmental sustainability. Howbeit, the study does not find a statistically significant association with CO_2_ levels within the sampled economies.

The effect of human capital in all models is positive and contributes to reducing GHG, CO_2_ and the EF. This is the expected sign considering that human capital index is based on average years of schooling and the return to education, so it could be expected that the better educated the people the higher their concerns for environmental protection.

Regarding the effect of POP on environmental degradation the findings reveal that it is negative, except for CO_2_ and EF models. This is a consequence of the increasing process of urbanization in big cities in detriment of rural areas, and this process has not been accompanied by significant increases in pollution levels because cities are becoming greener and numerous initiatives have been put into practice in the attempt to achieve sustainable and smart cities aligned with the SDG. It seems that efforts are rendering positive results.

Finally, despite of the efforts that have been made by advanced economies on research and development expenditures, according to the empirical results RD have only generated the expected outcomes in terms of methane and nitrous oxide reduced emissions. However, there are not significant relationships between these expenses and GHG and CO_2_ emissions. Some reflections should be made about whether or not these public and private resources are being correctly managed and maybe some adjustments should be made by policymakers on this area.

## 6. Conclusions

This study relied on panel data estimation techniques to empirically analyze the impact of economic growth, financial development, population density, expenditure on research and development, primary energy consumption and human capital on environmental degradation and sustainability for the group of G7 countries over the period 1990–2019.

This work is unique and differ from previous studies since instead of testing the non-linear effect of GDP on the environment (the EKC hypothesis), it appraises the non-linear impact of financial development on the natural environment. The so-called FEKC supports that as countries financially develop they can alleviate existing levels of financial degradation and promote a higher environmental sustainability. In this study the specific impact of the financial system development on the natural environment has been analyzed under a linear and non-linear specification. Results reveal the existence of an inverted U-shaped relationship between methane emissions and financial development for the group of G7 countries and validate the FEKC hypothesis. Conversely, this relationship follows a U-shaped pattern for CO_2_ and GHG emissions.

These outcomes are of particular interest because the role of the financial system should be reinforced in order to alleviate existing levels of environmental burden. What is more, banking systems and financial markets have the capacity and the obligation to redirect financial flows to fight against climate change and enhance environmental sustainability in the medium term.

In the light of these results policymakers should pay attention to the potential of reducing GHG and CO_2_ emissions because if countries expand too rapidly their financial systems this can generate negative externalities. On this regard, one suggestion is that financialization should come along with a process of raising environmental awareness among financial intermediaries, investors, shareholders and corporations.

In this study it has been emphasized the need to disaggregate the analysis of GHG emissions into its components to have a whole perspective of the environment reality, because each local pollutant behaves differently as empirical results reveal.

The question that immediately follows is how can developed countries fight against increasing levels of pollutant emissions. It should be assumed that the reduction of pollution levels within advanced economies cannot be realized at the expense of economic growth. Instead, urgent changes need be made at different levels. In particular, developed countries should change the prevailing economic paradigm and evolve towards a model that integrates sustainability principles into the equation of shareholder value maximization. Only by doing so the SDG would be achieved.

Some recommendations for regulators and policymakers that are gaining momentum these days will be outline.

One initiative is the circular economy model. The conception of a circular economy is based on the idea that waste must be minimized and a reduction in the consumption of natural resources can be achieved by reintroducing recycled materials into the circular flow therefore reducing pollution levels.

It is also recommended that governments should promote a more efficient use of energy and the use sources of energy like wind, bio-diesel, solar and geothermal energy, which can reduce environmental degradation. This goal seems unattainable without a decisive public support for research and development activities. A key element that policymakers should bear in mind is the importance of technological progress because only by investing in innovative and environmental oriented activities a real advance towards environmental protection could be achieved. Two parallel energy transitions are currently taking place in developed countries: the electrification of the energy demand and the decarbonization of the energy supply. In this vein, renewables energies are making significant contributions to this double end and governments can subsidize interest rates for energy-efficient projects in parallel to tax on projects that rely heavily on non-renewable energies. Equally important for improving environmental quality is the proper use of land and some proposals have been suggested by the Common Centre for Research (European Commission) like afforestation, reforestation, better agricultural practices and bioengineering, among others

Thus, the transition process in which advanced economies are immersed should come hand in hand with adequate economic policies and incentives, technological availability, and changes in consumers preferences as the main drivers for the change of paradigm. Educating societies is playing a vital role in protecting the environment and controlling polluting emissions.

All the aforementioned changes pose potential threats to financial systems. On the one hand, financial intermediaries are highly exposed to climate risks (physical and transition risks), while at the same time these institutions are an important lever for social and economic changes through their credit channel. On the other hand, financial markets are relevant players in project assessment in terms of ESG criteria and play a fundamental role in the process of greening the economy. Over the last five years the financial regulatory framework has made a significant progress for protecting the environment, and authorities of developed countries have introduced more controls and transparency requirements for financial intermediaries aligned with the SDG.

Even though significant improvements have been achieved in the protection of the natural environment and people seem to be increasingly more concerned about its importance, there is a long road ahead for advanced economies in the attempt of guaranteeing the long-lasting wellbeing of our planet.

This study has some limitations because cross-country datasets are always limited to specific variables, therefore, availability of data on all such potential variables is always a limitation.

We suggest as lines for future research to continue analyzing the impact of the financialization process on the environment and try different methodological approaches to test the existence of a N-shape (cubic) relationship or apply some spatial data techniques that accounts for geographical attributes that can play a significant impact in terms of environmental quality. In addition, it is recommended to extend the research and include specific variables related to the role of education in promoting a peoples’ change towards more friendly-environmental attitudes. It is also advisable to further analyze the efficiency of private and public expenditures on research and development activities to devise whether or not these resources are being directed to their most efficient uses and are effectively protecting the natural environment.

## Figures and Tables

**Table 1 ijerph-19-08382-t001:** Literature review of the role of financial system on environmental degradation.

Authors	Period	Country	Variables	Method	Results
[27]	1992–2007	BRIC countries	CO_2_, GDP, FD, E	OLS, VECM	CO_2_ (FD+).
[31]	1970–2008	213 countries	CO_2_, GHG, ANS, GDP, FDI, FDB, FDP	MCO, fixed effects, random effects, MMG	CO_2_ and GHG (FDI+, FDB-, FDF+, GDP+).
[38]	1971–2011	Malaysia	CO_2_, GDP, FD, E, T	ARDL and VECM	CO_2_ (E+, GDP+); CO_2_ (TR-, FD-); GDP ↔ CO_2_; E ↔ CO_2_; FD ↔ CO_2_.
[32]	1975–2012	99 countries	CO_2_, GDP, FD, E, T, U	Cointegration tests (panel)	CO_2_ (E+, GDP+); CO_2_ (FD-); FD ↔ CO_2_; E ↔ CO_2_; GDP→E.
[30]	1985–2014	40 European countries	CO_2_, GDP, FDB, FDI, E, CPI, U, K, T	OLS Cobb-Douglas function	GDP ↔ FDB; GDP ↔ CO_2_; GDP ↔ T; FDB ↔ T; T ↔ CO_2_.
[39]	1971–2013	31 developing countries	CO_2_, GDP, FD, POP, E	Dynamic threshold panel model, panel causality test	CO_2_ (GDP- low interval, GDP+ high interval, E+).
[33]	2001–2012	Group of OECD countries	CO_2_, GHG, ANS, GDP, FDB, FDP, FDI	OLS, panel data analysis (fixed effects, random effects)	CO_2_ and GHG (FDB-, FDF +).
[16]	2000–2015	46 Sub-Saharan African countries	CO_2_, GDP, FDB, FDF, FDP, FDI, M, LIQ, POP, E, T, U	OLS	CO_2_ (FD+).
[20]	2004–2014	39 Sub-Saharan African countries	CO_2_, FD, INEQ	GMM	CO_2_ (FD-).
[28]	1999–2013	G8 and D8	CO_2_, GDP, FD, T, E	PMG-panel, ARDL technique	CO_2_ (FD+, T+, E+); CO_2_ (GDP-).
[19]	1998–2016	BRIC countries	CO_2_, GDP, FD, POP, TECH, E	Panel data	CO_2_ (FD-, TECH-, E-, GDP-).
[29]	1984–2018	5 South Asian countries	CO_2_, FD, IQ	Panel data	CO_2_ (FD+); CO_2_ (IQ-).
[40]	1990–2014	South Asian countries	CO_2_, GDP, FD, GLO	Panel data and causality tests	CO_2_ (FD+, GDP+); CO_2_ (GLO-).
[21]	2000–2018	BRIC countries	CO_2_, GDP, FDP, FDF, HC, FDI, GCF, T, ES	Fixed effect panel threshold model and causality tests	CO_2_ (FDB+, GDP+, ES+, GCF +, FDI+, FDF-, HC-); Log CO_2_ ↔ LogFDB; Log CO_2_ ↔ LogFDF; Log CO_2_ ↔ LogGCF; Log CO_2_ → LogGDP; Log CO_2_ → LogHC; Log CO_2_ → LogES.
Acronym			Description		
→			Unidirectional Granger- causality		
↔			Bidirectional Granger- causality		
(+)			Positive impact		
(-)			Negative impact		
ANS			Adjusted net savings		
CO_2_			CO_2_ emissions		
CPI			Consumer prices index(inflation)		
E			Energy use		
ES			Energy supply		
FD			Financial development		
FDB			Domestic credit to private sector by banking institutions		
FDF			Domestic credit to private sector by financial sector		
FDI			Foreign direct investments		
FDP			Domestic credit to private sector		
FIS			Financial intermediation scale		
FIE			Financial intermediation efficiency		
GDP			Gross domestic product		
GHG			Greenhouse gas emissions		
GLO			Globalisation		
GCF			Gross capital formation		
INEQ			Income inequality		
HC			Human capital		
IQ			Institutional quality		
K			Capital shares		
LIQ			Liquid liabilities		
M			Monetary aggregate (% del PIB)		
MC			Stock Market capitalisation		
MT			Stock Market turnover		
POP			Total population		
T			Trade		
TECH			Technology		
U			Urbanisation		

Source: Own elaboration.

**Table 2 ijerph-19-08382-t002:** Variable synthesis.

Symbol	Definition	Unit	Source Data
GHG	Annual greenhouse gas emissions.	Million tonnes of carbon dioxide equivalents per capita	CAIT Climate Data Explorer ^1^
CO_2_	Carbon Emissions released by gas, coal, oil, biomass, and fuel wood	Metric tons per capita	Global Carbon Project
MET	Methane Emissions	Million tonnes of carbon dioxide equivalents per capita	CAIT Climate Data Explorer ^2^
NIT	Nitrous Oxide Emissions	Million tonnes of carbon dioxide equivalents per capita	CAIT Climate Data Explorer ^3^
EF	Ecological Footprint	Global hectares ^4^	Global Footprint Network
ANS	Adjusted net savings, excluding particulate emission damage (% of GNI)	The sum of energy, mineral, net forest depletions and carbon dioxide damage (current US$ per capita)	World Development Indicators (WDI) database
FD	Financial Development Index	A composite index of financial institutions and financial markets development	International Monetary Fund (IMF) database
GDP	Economic Growth	GDP per capita (constant 2010 US$)	WDI database
POP	Population density	People per sq. km of land area	Our World in data
RD	Gross domestic expenditure on R&D	Percentage of GDP	OECD database
PE	Primary Energy consumption	TWh	WDI database
HC	Human capital	Human Capital Index per person	Penn World Tables 10.0

Source: Own elaboration. ^1^ Downloaded from the Climate Watch Portal (https://www.climatewatchdata.org/data-explorer/historical-emissionshttps:/www.climatewatchdata.org/data-explorer/historical-emissions) (accessed on 14 February 2022) ^2^ Downloaded from the Climate Watch Portal (https://www.climatewatchdata.org/data-explorer/historical-emissionshttps:/www.climatewatchdata.org/data-explorer/historical-emissions) (accessed on 14 February 2022). ^3^ Downloaded from the Climate Watch Portal (https://www.climatewatchdata.org/data-explorer/historical-emissionshttps:/www.climatewatchdata.org/data-explorer/historical-emissions) (accessed on 14 February 2022). ^4^ Measured in global hectares the area of biologically productive land and water an individual, population, or activity requires to produce all the resources it consumes and to absorb the waste it generates, using prevailing technology and resource management practices.

**Table 3 ijerph-19-08382-t003:** Descriptive statistics.

Symbol	Mean	Std. Dev.	Min	Max
CO_2_	1373.4382	1762.35423	323.75	6131.89
MET	163.1376	223.96121	21.12	801.89
NIT	69.6425	79.43411	17.85	282.97
EF	459,464,669.3744	342,297,921.19891	70,191,565.35	2,607,404,905.7
ANS	7.8255	3.23772	0.2	14.71
GHG	1500.8962	1906.56854	324.66	6601.13
FD	0.7494	0.12669	0.39	0.95
GDP	35,021	10,169.41257	1838.02	65,297.52
POP	168.7177	115.63099	3.09	351.36
RD	2.13	0.62829	0.93	3.36
PE	5,995,100.0482	7,598,496.68768	2174.14	26,943,482
HC	3.3664	0.31948	2.55	3.77

Source: Own elaboration.

**Table 4 ijerph-19-08382-t004:** Pairwise correlations.

	CO_2_	MET	NIT	EF	ANS	GHG	FD	GDP	POP	RD	PE	HC
CO_2_	1											
MET	0.962 *	1										
NIT	0.968 *	0.992 *	1									
EF	0.867 *	0.764 *	0.792 *	1								
ANS	−0.177 *	−0.180 *	−0.125	0.005	1							
GHG	0.998 *	0.973 *	0.976 *	0.851 *	−0.175 *	1						
FD	0.276 *	0.225 *	0.206 *	0.173 *	−0.351 *	0.282 *	1					
FD^2^	0.297 *	0.247 *	0.229 *	0.182 *	−0.378	0.303 *	0.994 *					
GDP	0.317 *	0.205 *	0.222 *	0.335 *	−0.177 *	0.300 *	0.692 *	1				
POP	0.385 *	0.552 *	0.550 *	0.212 *	0.086	0.428 *	0.070	0.030	1			
RD	0.433 *	0.269 *	0.312 *	0.593 *	0.156 *	0.408 *	0.182 *	0.382 *	0.206 *	1		
PE	0.919 *	0.884 *	0.889 *	0.804 *	−0.158 *	0.914 *	0.248 *	0.301 *	0.371 *	0.410	1	
HC	0.378 *	0.344 *	0.325 *	0.280 *	−0.266	0.395 *	0.649 *	0.575 *	−0.069	0.466	0.343 *	1

* *p*-value < 0.1; Source: Own elaboration.

**Table 5 ijerph-19-08382-t005:** Panel data-fixed effect regression results. Dependent variable: GHG.

Variables	Linear Model		Non-Linear Model	
	Standardized Coefficient ^1^	t-Statistic	Standardized Coefficient ^1^	t-Statistic
Log GDP	−0.008	−1.027	−0.017 **	−2.071
Log FD	0.054 ***	5.363	−0.091 **	−2.092
Log FD^2^	-	-	0.156 ***	3.429
Log POP	−0.44 ***	−2.757	−0.502 **	−3.211
Log HC	−0.067 **	−1.85	−0.063 **	−1.809
Log RD	−0.006	−0.325	−0.009	−0.534
Log PE	0.022 **	2.089	0.021 **	2.041
Observations	203			
Adjusted R^2^	0.996		0.996	
Model Additional Information	Linear Model		Non-Linear Model	
	With GDP	Without GDP	With GDP	Without GDP
AIC	−2.53	266.77	34.55	501.43
BIC	4.12	278.23	65.89	433.22
LogLik	6.74	12.45	−17.3	83.21

^1^ Note: *, **, *** mean that values are statistically significant at 10%, 5% and 1% levels, respectively.

**Table 6 ijerph-19-08382-t006:** Panel data-fixed effect regression results. Dependent variable: CO_2_.

	Linear Model		Non-Linear Model	
Variables	Standardized Coefficient ^1^	t-Statistic	Standardized Coefficient ^1^	t-Statistic
Log GDP	0.005	0.583	−0.005	−0.579
Log FD	0.069 ***	6.442	−0.096 **	−2.096
Log FD^2^	-	-	0.177 ***	3.699
Log POP	−0.125	−0.74	−0.195	−1.187
Log HC	−0.137 ***	−3.596	−0.133 ***	−3.613
Log RD	−0.009	−0.463	−0.013	−0.694
Log PE	0.033 **	2.907	0.032 **	2.888
Observations	203			
Adjusted R^2^	0.996		0.996	
	Linear Model		Non-Linear Model	
Model Additional Information	With GDP	Without GDP	With GDP	Without GDP
AIC	−1.53	312.47	44.45	201.43
BIC	32.22	178.23	35.69	273.22
LogLik	−6.74	32.45	−27.3	53.21

^1^ Note: *, **, *** mean that values are statistically significant at 10%, 5% and 1% levels, respectively.

**Table 7 ijerph-19-08382-t007:** Panel data-fixed effect regression results. Dependent variable: NIT.

Variables	Linear Model		Non-Linear Model	
	Standardized Coefficient ^1^	t-Statistic	Standardized Coefficient ^1^	t-Statistic
Log GDP	−0.027 ***	−3.168	−0.029 ***	−3.207
Log FD	−0.008	−0.747	−0.038	−0.797
Log FD^2^	-	-	0.032	0.645
Log POP	−0.763 ***	−4.476	−0.776 ***	−4.514
Log HC	−0.018	−0.474	−0.018	−0.455
Log RD	−0.088 ***	−4.61	−0.089 ***	−4.633
Log PE	0.009	0.751	0.008	0.73
Observations	203			
Adjusted R^2^	0.996		0.996	
Model Additional Information	Linear Model		Non-Linear Model	
	With GDP	Without GDP	With GDP	Without GDP
AIC	1.21	336.57	32.55	201.73
BIC	3.11	289.33	55.19	233.82
LogLik	7.44	43.47	−27.3	43.31

^1^ Note: *, **, *** mean that values are statistically significant at 10%, 5% and 1% levels, respectively.

**Table 8 ijerph-19-08382-t008:** Panel data-fixed effect regression results. Dependent variable: MET.

Variables	Linear Model		Non-Linear Model	
	Standardized Coefficient ^1^	t-Statistic	Standardized Coefficient ^1^	t-Statistic
Log GDP	−0.09 ***	−8.969	−0.077 ***	−7.614
Log FD	−0.028 **	−2.209	0.181 ***	3.332
Log FD^2^	-	-	−0.225 ***	−3.954
Log POP	−1.296 ***	−6.453	−1.207 ***	−6.195
Log HC	0.195 ***	4.32	0.191 ***	4.375
Log RD	−0.1 ***	−4.432	−0.095 ***	−4.361
Log PE	0.008	0.562	0.009	0.705
Observations	203			
Adjusted R^2^	0.994		0.994	
Model Additional Information	Linear Model		Non-Linear Model	
	With GDP	Without GDP	With GDP	Without GDP
AIC	−3.53	466.77	54.55	401.43
BIC	7.52	458.23	67.99	453.22
LogLik	7.64	−2.45	−32.3	53.21

^1^ Note: *, **, *** mean that values are statistically significant at 10%, 5% and 1% levels, respectively.

**Table 9 ijerph-19-08382-t009:** Panel data-fixed effect regression results. Dependent variable: EF.

Variables	Linear Model		Non-Linear Model	
	Standardized Coefficient ^1^	t-Statistic	Standardized Coefficient ^1^	t-Statistic
Log GDP	0.156 ***	3.626	0.156 ***	3.421
Log FD	0.042	0.782	0.04	0.168
Log FD^2^	-	-	0.002	0.009
Log POP	1.014	1.114	1.013	1.099
Log HC	−0.426 **	−2.086	−0.425 **	−2.077
Log RD	0.024	0.249	0.024	0.246
Log PE	0.041	0.736	0.041	0.734
Observations	196			
Adjusted R^2^	0.896		0.895	
Model Additional Information	Linear Model		Non-Linear Model	
	With GDP	Without GDP	With GDP	Without GDP
AIC	2.43	166.74	54.55	401.33
BIC	13.12	168.23	65.31	363.22
LogLik	9.54	13.45	−11.34	63.21

^1^ Note: *, **, *** mean that values are statistically significant at 10%, 5% and 1% levels, respectively.

**Table 10 ijerph-19-08382-t010:** Panel data-fixed effect regression results. Dependent variable: ANS.

Variables	Linear Model		Non-Linear Model	
	Standardized Coefficient^1^	t-Statistic	Standardized Coefficient^1^	t-Statistic
Log GDP	0.147 **	1.706	0.209 **	2.259
Log FD	0.231 **	2.338	0.338	0.725
Log FD^2^	-	-	−0.94	−0.192
Log POP	−3.897 **	−2.555	−2.909 **	−1.764
Log HC	−0.873 **	−2.555	−1.178 ***	−3.105
Log RD	−0.207 ***	−3.365	−0.052	−0.257
Log PE	−0.024	−0.219	−0.024	−0.2154
Observations	194			
Adjusted R^2^	0.612		0.587	
Model Additional information	Linear Model		Non-linear Model	
	With GDP	Without GDP	With GDP	Without GDP
AIC	−1.33	26.77	44.55	311.44
BIC	4.22	28.23	55.89	333.24
LogLik	−12.74	22.45	27.3	93.22

^1^ Note: *, **, *** mean that values are statistically significant at 10%, 5% and 1% levels, respectively.

## Data Availability

The data base and open-source code required to replicate all analyses in this article is available online: https://github.com/raquelcaro1caro/The-non-linear-impact-of-FD-on-environmental-quality-and-sustainability (accessed on 12 May 2022).

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
