# Peer review of "The Non-Linear Impact of Financial Development on Environmental Quality and Sustainability: Evidence from G7 Countries"

_ijerph, 2022, doi:10.3390/ijerph19148382_

Round 1
Reviewer 1 Report
This paper tries to evaluate whether the financial development of OECD countries implies a reduction of several environmental indicators. The conceptual framework is the one of the kuznets curve that has been widely explored in literature.
The added value of the paper is residing in the fact that the authors use the financial development instead that using the standard GDP per capita as independent variable.
Results demonstrate that the inverted u-shaped curve is not empirically validated with the exception of the case of methane emissions.
My sensation is that the paper presents some weaknesses. First, I would like to see what happens when you exclude the GDP from the regression. The GDP and the financial development are highly correlated and moreover, the use of GDP is, per se, a testo of the standard EKC.
Moreover, I suspect that the financial development is linked with the increase of the role of services in the economy of a country. This means that developed country, with a well developed financial systems, rely on services, while reducing the role of polluting industry within the country. What happens is that the manufacturing and industry, which are highly polluting are just placed abroad (leakage effect). So, it is stil more surprising to see that we have a U-shaped curve. Should you test for the N pattern? Thus, to address the aspect related with the leakage effect, maybe you can include some variable of trade and openness with countries with a lower environmental regulation than EU o USA.
Moreover, why the authors use only 7 OECD countries and not all the countries? this is not clear, since for what is my concern there are data for all the oecd countries. I would extend the analysis.
Before taking a decision , I think that these analysis should be presented or authors could provide rebuttals for the above comments.
Reviewer 2 Report
The connection between finance and the environment is an important research niche. The analysis of non-linear relationship between financial system development and the environment, that the authors perform, is indeed novel in the way that it is analysed in this article and in terms of methodology used.
The methodology itself is appropriate and in line with the state-of-the-art in the field. Paper structure is appropriate and the arguments flow clearly and convincingly. A vast array of carefully selected sources is used by the authors.
The conclusions are both interesting, go beyond state-of-the-art and could be used for further research.
For all of the above reasons, I support the publication of the article in the present form.
Reviewer 3 Report
Dear Editors,
Have a nice day!
This paper analysis the impact of financial development on the environmental quality of G7 countries. I found the introduction, methods, analysis, and results very clear and technically sound.
This paper analysis the non-linear impact of financial development on environmental quality with reference to G7 countries. The uniqueness of this paper as mentioned in line 101 is the analyzing the non-linear relationship between financial system development and the natural environment withing the EKC framework. However, this unique aspect could be more justified with some citations following the research gap.
Introduction:
In the abstract, the authors mention that they define a new Financial Environmental Kuznets Curve (FEKC) by introducing the square term of financial development on the environment-finance relationship. However, the introductory part misses the related information that what should be expected to read in the paper related to FEKC.
Literature review:
The literature review provides sufficient information related to EKC with other studies and authors’ FEKC viewpoint.
Data and results:
Time series data are used from 1990 to 2019 to compare the linear and non-linear models which confirm that FD is non-statistically significant.
Results and conclusion:
The results are clearly described serving the agenda for this research and conclusions providing policy implication and future directions for research. At this point, as EKC informs education as the turning point for financially self-sufficient countries, the authors could further suggest to extend this research with educational variables.
Finally, this paper contributes to the knowledge related to the area and is technically solid. It catches the attention as a reader. Therefore, I feel this is good for publication.
Round 2
Reviewer 1 Report
The authors provide suitable replies to my comments. The shortcomings of the paper are described in the text and this should guarantee the clarity of the interpretation of the results. The paper can be accepted in its current form.